# Interventions for School-Aged Children with Auditory Processing Disorder: A Scoping Review

**DOI:** 10.3390/healthcare12121161

**Published:** 2024-06-07

**Authors:** Jacynthe Bigras, Josée Lagacé, Ahmed El Mawazini, Héloïse Lessard-Dostie

**Affiliations:** School of Rehabilitation Sciences, Faculty of Health Sciences, University of Ottawa, Ottawa, ON K1N 6N5, Canada; josee.lagace@uottawa.ca (J.L.); aelma034@uottawa.ca (A.E.M.); hlessar2@uottawa.ca (H.L.-D.)

**Keywords:** auditory processing disorder, speech in noise, intervention, school-age, outcome, scoping review

## Abstract

(1) Background: Auditory processing (AP) disorder is associated with learning difficulties and poses challenges to school-aged children in their daily activities. This scoping review identifies interventions and provides audiologists with protocol insights and outcome measures. (2) Methods: A systematic search of both peer-reviewed and grey literature (January 2006 to August 2023) covered ten databases. Studies included had the following characteristics: (i) published in French or English; (ii) participants were school-aged, and had a normal audiogram, AP difficulties or disorder, and no cognitive, developmental, congenital or neurological disorder (with the exception of learning, attention, and language disabilities); (iii) were intervention studies or systematic reviews. (3) Results: Forty-two studies were included, and they predominantly featured auditory training (AT), addressing spatial processing, dichotic listening, temporal processing and listening to speech in noise. Some interventions included cognitive or language training, assistive devices or hearing aids. Outcome measures listed included electrophysiological, AP, cognitive and language measures and questionnaires addressed to parents, teachers or the participants. (4) Conclusions: Most interventions focused on bottom-up approaches, particularly AT. A limited number of top-down approaches were observed. The compiled tools underscore the need for research on metric responsiveness and point to the inadequate consideration given to understanding how children perceive change.

## 1. Introduction

Learning and communication require listening skills, especially in the classroom and when interacting with peers, teachers, community members and family [1,2]. School-aged children spend a significant amount of time in educational environments, where classroom acoustic conditions can be challenging. Indeed, the characteristics of background noise, such as the spectrum of the noise and the fluctuations in the noise and the reverberations, may interfere with the voice of the teacher and those of the other students [1], necessitating increased listening effort [3,4]. Howard et al. (2010) found that school-aged children with normal hearing (NH) and without language, learning, or cognitive disorders required considerable listening effort when performing tasks in typical classroom signal-to-noise ratios (SNRs). They suggest that fewer cognitive resources were available for additional tasks as a result [3]. While classrooms are demanding listening environments for neurotypical children, they can pose even greater challenges for children with learning difficulties, as these children often struggle to listen to speech, particularly in background noise [5,6,7].

The nature of listening difficulties in school-aged children can be multifactorial, and may be caused by overlapping deficits in hearing, cognition, language or auditory processing (AP) [8]. Furthermore, the symptoms of these deficits may intersect, making the identification of the underlying deficits complex [8,9,10]. Auditory processing disorder (APD) can be the underlying cause of the listening difficulties, particularly in cases where the children have NH thresholds and do not present with any known language or cognitive deficits. Nevertheless, for many children with NH thresholds, the nature of their listening difficulties is less clear, as they present with AP challenges and one or more concomitant conditions such as dyslexia [6,11] or difficulty with cognitive skills, including attention or memory [12,13]. Children who have APD and AP difficulties comprise a heterogeneous population [2]. However, many of these children commonly experience difficulties, such as academic challenges and struggling to perceive speech or difficulties in concentration when there is noise, when following complex directions, understanding instructions, interpreting messages correctly and answering questions within a time frame, as compared to children of the same age [6,12,14,15]. Furthermore, listening difficulty in background noise is common in these children, despite having NH thresholds. Purdy et al. (2018) found that children diagnosed with an APD and those who reported listening difficulties, even in the absence of APD, experienced more hurdles in classroom listening situations compared to a control group of children without such difficulties or APD [2].

Although there are several publications, including professional guidelines and position statements, on the assessment of APD, no agreement on the definition or the ideal test battery has yet been reached [8,9,16,17]. AP refers to the central auditory nervous system’s ability to “preserve, refine, analyze, modify, organize, and interpret” [18] information received from peripheral auditory structures [18]. This ability requires various essential AP mechanisms, including localizing/lateralizing sounds; listening to degraded speech or speech in a noisy environment (e.g., in a restaurant or on an airplane); recognizing the temporal aspects of speech (e.g., sound order, intonation and short silences); and auditory discrimination and auditory patterns [18,19]. APD originates in the auditory system and is characterized by deficits in AP mechanisms [15,18,19]. APD often occurs alongside other disorders, such as attention deficit and hyperactivity disorder (ADHD), language impairment and learning disabilities. This makes diagnosing APD more complex and highlights the need for a multidisciplinary approach, a thorough case history, and further research with scientific rigor to better distinguish APD from other neurodevelopmental disorders or to understand their similarities and differences better [10,20]. The diagnosis of APD is often based on performance below the mean (in at least one ear at two standard deviations or more) on a minimum of two of the AP tests of the relevant battery [15,18]. The AP test battery typically comprises psychoacoustics tests with verbal and non-verbal stimuli, but objective audiological measures are recommended, such as auditory brainstem response (ABR), acoustic reflex thresholds and otoacoustic emissions [10]. In the case of children, AP difficulties are often developmental [17]. Therefore, a re-evaluation is recommended two years following the first assessment to confirm the presence of an APD [15]. Some children may show improvement in their test results over time, indicating that there could have been a delay in the maturation of their AP abilities. The current review will focus on two groups of children: those with an APD based on the previously mentioned guidelines, and those who have AP difficulties identified through abnormal results in one or more AP tests. The latter group includes children who have not completed a full APD test battery or who have completed a standard APD test battery once.

The existing clinical practice guidelines in audiology provide recommendations for intervening with children who have APD or AP difficulties [15,18,21,22]. Canadian guidelines aim at improving participation in daily life-activities and use the International Classification of Functioning, Disability and Health model created by the World Health Organization to guide the interventions [15,23]. These guidelines suggest that management of APD should focus on personal factors, such as direct auditory training (AT) and communication skills training, and on environmental factors, such as the acoustics of the environment and the use of assistive devices [15]. The recommendations of the British and the American practice guidelines focus on improving the environmental listening conditions, direct AT and compensatory strategies [18,21].

Several published review articles have aimed to identify and synthesize intervention strategies for various populations facing challenges in one or more AP abilities. However, the interventions listed do not provide a comprehensive perspective, making it challenging for clinicians to identify suitable interventions for their clients, as well as the appropriate outcome measures. Below, summaries of some of the review articles are provided.

Bamiou et al. (2006) reviewed management strategies available for children with APD. Although they concluded that more research is needed to understand the impact of interventions, their review highlighted some interventions that have the potential of improving, in particular, auditory evoked potentials (AEPs), AP test scores, phonological awareness and academic performance [24]. The specific intervention strategies listed by Bamiou et al. (2006) focus on environmental modifications, assistive listening devices, teacher- or speaker-based adaptation, formal and informal AT and compensatory strategies [24]. The formal AT programs discussed include computer-based training. Informal AT is also described and includes auditory closure activities; temporal patterning and prosody training, such as recognition and use of prosodic aspects of speech (rhythm, stress and intonation); auditory discrimination training, such as phonological awareness; dichotic listening training; and speech-in-noise (SIN) training. The review includes a list of compensatory strategies that can be used to remediate listening difficulties, such as active listening, auditory memory enhancement and metacognitive and metalinguistic strategies.

In 2011, Fey and colleagues conducted a systematic review focusing on peer-reviewed studies that explored the impact of auditory or language interventions on school-age children diagnosed with an APD and/or a spoken language disorder [25]. Six intervention studies on children with an APD were included. Interventions listed included traditional computer-based auditory and language training, dichotic listening training, and listening in competing noise. The authors noted that most studies lack blinding of the testers to the participants’ group, and that the participants were not randomly assigned to the experimental groups. In summary, based on the systematic review, the authors did not find convincing evidence supporting the notion that auditory interventions enhance auditory, language, or academic outcomes in school-age children with an APD or a language disorder [25]. In response to the systematic review by Fey et al. (2011), Bellis and colleagues (2012) argued that Fey et al.’s review employed a broad definition of APD and that the interventions listed were not AT approaches recommended by current APD guidelines, noting that some focused on spoken and written language [25,26]. They contended that the search criteria were too selective, such as the age range of the population searched, resulting in the exclusion of relevant research. Bellis and colleagues (2012) referenced several studies excluded from Fey et al.’s (2011) review [25,26]. These studies highlighted improvements, including, but not limited to, neurophysiological outcome measures that support AT for children with an APD. Some of the studies cited by Bellis et al. (2012) were, however, published after the search dates of Fey et al.’s (2011) review [25,26]. Bellis et al. concluded their response by stating their agreement with Fey et al. (2011) regarding the necessity for additional research in this particular field [25,26]. Fey et al. (2012) responded and argued that their methodology was well-based and that their conclusions were still valid [27].

In their review article, Weihing et al. (2015) described AT procedures with children and adults who have an APD. They reiterated the importance of including different approaches in the management of APD, including environmental modifications, assistive devices, compensatory strategies and direct AT [28]. Additionally, their review includes the detailed paradigms of the training listed, as well as the characteristics and parameters of the training. They discuss the use of computer-based training programs, such as auditory and auditory-language software. These training strategies focus on dichotic listening, spatial processing, listening to SIN, phonological awareness, auditory discrimination and temporal processing. In the review, they describe practical details as to training schedules, training difficulty and transfer of learning skills. Overall, some of the children with an APD who completed temporal processing or auditory discrimination training showed improvements in phonemic synthesis, temporal processing and language and reading outcomes, but improvement was also seen in cognition abilities (attention) and electrophysiological auditory components. Following dichotic training, children had a significant increase in left-ear performance in a dichotic digits test. Children who completed a spatial processing training were better at using spatial cues in noise. Improvement in attention and memory and as to hearing handicap measures were also observed [28].

More recently, Gohari et al. (2022) described, in their review, diverse strategies for improving speech perception in noise. Their review primarily examined various populations, but they noted studied interventions with school-aged children with APDs. These interventions included localization training, listening in spatialized noise training and SIN training, with a variety of speech material in specific SNRs [29]. Overall, improvements were observed across various aspects, including binaural processing abilities, noise tolerance, electrophysiological components, and behavioural measures of AP, such as speech perception in noise and temporal processing [29].

The clinical guidelines and the review studies presented above all conclude that research is needed to develop evidence-based rehabilitation strategies for APD [15,21,22,24,25,28]. The need to develop outcome measures for documenting the effectiveness of rehabilitation programs is also mentioned [15]. In recent years, several interventional studies focusing specifically on school-aged children with an APD or AP difficulties have been published, creating a need to synthesize the information and to make it more accessible to clinicians and researchers. The first objective of the current scoping review was to systematically map, compile and synthesize the studies exploring interventions for school-aged children with NH thresholds and an APD or AP difficulties. This includes, among other aspects, describing the types of protocols and parameters of the interventions investigated. A second objective was to list the outcome measures used to document the effectiveness of the interventions. This scoping review will facilitate the summary and dissemination of the research findings [30] and will contribute to the knowledge of clinicians regarding available interventions and outcome measures for a population commonly encountered in clinical settings.

## 2. Materials and Methods

### 2.1. Protocol and Registration

Each stage of development for this scoping review (registration osf.io/4gn6a) followed the protocol suggested by the Preferred Reporting Items for Systematic Reviews and Meta-Analysis extension for scoping reviews (PRISMA-ScR) [31].

### 2.2. Eligibility Criteria

To be included in the scoping review, studies had to focus on interventions for improving AP in children primarily aged between 6 and 12 years old. Studies also had to report outcome measures. Eligibility of the participants in the studies was based on a diagnosis of an APD or abnormal results at AP tests (e.g., dichotic listening, SIN perception, degraded speech and temporal processing). Peer-reviewed journal papers were included if they were published in 2006 or later, written in French or in English (languages spoken by the authors), and were either intervention studies, systematic reviews, meta-analyses, or scoping reviews. To provide a comprehensive overview of the topic, the grey literature was also searched using the same eligibility criteria. For the grey literature, included studies had to be complete, and be freely accessible or accessible through the University of Ottawa (UO) library. The grey literature included master’s or doctoral theses. The chosen period was deemed appropriate to ensure a manageable volume of literature while providing the most recent intervention protocols.

Studies were excluded if they focused on participants with hearing loss, traumatic brain injury, or neurological, cognitive, developmental, or congenital disorders. Because learning disabilities, such as reading disorders, ADHD and/or speech-language impairment are often concomitant with AP challenges [6,7], studies were included if participants had those particular diagnoses. Studies completed on animals, studies not available in full-text, patient reports, and descriptions of medical treatments such as surgeries were excluded.

### 2.3. Information Sources

To identify potentially relevant studies, the following ten databases were searched from January 2006 to August 2023: Medline (Ovid), APA PsycInfo (Ovid), ERIC (Ovid), CINHAL (Ebsco), Education Source (Ebsco), Communication Source (Ebsco), Scopus and Scopus Conference Proceedings, Web of Science and Web of Science proceedings papers. As per guidance provided by a librarian at the UO, these databases were identified as the sources most likely to yield studies relevant to the review’s objective. Sources of grey literature were identified through Grey Matters (2019) [32] and using the website of the library of UO. ProQuest Dissertations and Thesis Global, speechBITE, National Institute for Health and Care Excellence (NICE), University of York Centre for Reviews and Dissemination, PROSPERO, ISRCTN, CenterWatch Clinical Trials Listing Service, ClinicalTrials.gov, Trip PRO and Google Scholar (first 100 hits [32]) were searched. Following the full-text screening, the first author (JB) conducted a thorough scan of the references in the included studies to identify any additional relevant research.

### 2.4. Search Strategy

The search strategy and terminology were chosen with the support of the librarian at the UO. Following the PRISMA-ScR protocol, the search strategy for at least one database is presented below [31]. Table 1 outlines the search strategy for Medline (Ovid). Citations retrieved from each database search were imported into Covidence [33], where duplicates were subsequently removed.

### 2.5. Selection of Sources of Evidence

Title and abstract screening were conducted independently by the first and second authors (JB and JL), using the Covidence platform [33]. Each article was meticulously reviewed for inclusion, with assessments made in accordance with the predefined inclusion and exclusion criteria. JB and JL conducted a practice round with 75 articles and practiced resolving disagreements, ensuring a consistent screening process. After the practice round, JB and JL screened the remaining articles based on the title and abstract. Any disagreements in study selection were resolved through consensus and discussion between the two authors. Full-text screening was completed by JB.

### 2.6. Data Charting

A data-charting table was developed in Excel and tested by JB and JL, based on the inclusion criteria and relevant data items necessary to address the aims of the scoping review. These authors tested the table with two different studies. The data-charting table was filled out independently by the two authors, and descriptive data was compared and checked for agreement. Any disagreements were resolved through consensus and discussion, and the data-charting table was updated. After testing the data-charting table, the finalized version was incorporated into Covidence [33] to allow for the systematic charting of the included studies’ the data. JB independently completed the data charting, and to ensure accuracy, AEM and HLD verified the data from these studies. Discrepancies were discussed and resolved when necessary.

### 2.7. Data Items

The charting table systematically captured pertinent information about the therapies, including the characteristics and the schedule of the interventions, the outcome measures, and the effectiveness of the interventions, including effect size. Data items for charting included study characteristics (i.e., year of publication, country where the study was conducted and study design); aim of the study; attributes of both the experimental and, if applicable, control groups (i.e., number of groups, number of participants, mean age, diagnosis and inclusion and exclusion criteria); intervention details (i.e., description, focus and provider of the intervention); parameters or schedule of the intervention (e.g., timetable, total duration of training, duration of each training session, number of training sessions and intervals between them); outcome measures (including time of measurement); main results (including whether changes were noted in the control group where applicable); and effect size when applicable. Furthermore, since the primary goal of most rehabilitation programs is to enhance participation in daily activities and ensure that the skills acquired in a controlled setting can be transferred to real-life situations [34] or to other untrained skills [28], the included studies were also examined for information on generalization where relevant. When evaluating the effectiveness of an intervention, incorporating generalization measures can contribute to the external validity of the intervention studied [35]. Outcome measures related to generalization of the results were also charted whenever they were explored.

### 2.8. Critical Appraisal

No critical appraisal of the included sources of evidence was conducted, as the primary objective of the scoping review was to provide an overview of the existing evidence [31]. Evaluating the strengths and weaknesses of each study was considered beyond the scope of the review.

### 2.9. Synthesis of Results

Characteristics of the studies and the participants are presented in a table found in the Appendix A. When reviews were identified, data were charted for the studies that met the inclusion criteria.

Compiling information on the measures used to evaluate the effectiveness of interventions was a key objective of this scoping review. Outcome measures are presented in Table A1 of Appendix B and are categorized into three groups: behavioural measures, neurophysiological measures and questionnaires. In this review, behavioural measures refer to tests that assess auditory, language, cognitive or academic abilities by observing participants’ performance and responses in various tasks. Neurophysiological measures assess the neural responses at different levels of the auditory nervous system. Finally, questionnaires are tools used to gather information about an individual’s experiences or opinions in a specific area or topic.

Details of the interventions, as well as the training schedules, are presented in Table A2 of Appendix C. To effectively address the objectives of the scoping review, the studies were categorized according to the method of intervention employed.
**Single-ability auditory training.** The first method described in this review is AT, which focuses on a single auditory ability and includes structured listening activities tailored to enhance that specific auditory mechanism [15,28,36]. The fundamental aim of AT is to improve the neural processing of acoustic signals.**Multiple-ability auditory training.** The second method is AT targeting multiple abilities. Just like single-ability AT, this approach includes structured listening activities tailored to enhance specific auditory mechanisms [15,28,36] and to improve the neural processing of acoustic signals.**Assistive devices (FM system)**. The use of the frequency modulation (FM) system, an assistive device, was another intervention method reported in the literature. According to the World Health Organization, assistive devices are products or piece of equipment that help maintain or improve a person’s functioning [23]. In the current review, the FM system was used in the classroom with the aim of enhancing the teacher’s voice and improving the student’s listening in the classroom.**Hearing aids.** These personal amplification devices, in which the receiver and microphone are both worn by the person who has difficulty hearing, can improve access to speech and environmental sounds and may improve SNRs for the person wearing the devices.**Top-down interventions.** These interventions focus on enhancing higher-order cognitive abilities (e.g., memory, attention, or problem-solving abilities), as well as language skills (e.g., vocabulary or metalinguistic abilities). Top-down interventions emphasize context and analysis of lower-order information, such as speech sounds. The aim is to improve listening abilities through compensatory and accommodative approaches [37].**Multimodal interventions**. These interventions incorporate a range of strategies aimed at enhancing AP abilities, listening skills and language skills and/or optimizing the listening environment. Multimodal interventions may involve a combination of bottom-up and top-down approaches addressing both fundamental auditory processes and higher-level cognitive functions. Additionally, multimodal interventions may include adaptations of the environment or the use of assistive devices with the goal of improving the listening experience.

## 3. Results

### 3.1. Selection of Sources of Evidence

Figure 1 illustrates the step-by-step process employed to determine the eligibility of articles for the scoping review. After eliminating duplicate entries, a total of 3083 studies were identified, originating in both the databases and the grey literature. Subsequently, these studies underwent screening based on the title and abstract, resulting in the exclusion of 2913 studies. A total of 170 full-text articles were assessed for eligibility, resulting in the exclusion of 128 articles. The exclusion of these articles was based on the following criteria: duplicate, not a study, inappropriate studied population (e.g., adults or absence of AP measures for participant identification), non-peer-reviewed status (if not a thesis), incorrect study design, chapter from a book, review or expert opinion, full text not available, study not written in French or English, absence of reported therapy addressing AP difficulties, or insufficient information on participants or intervention. The references of the 42 remaining articles were examined to identify any citations that were not initially retrieved from the databases and grey literature. No additional references were found in this process. Forty-two studies were included in the review.

### 3.2. Characteristics of Included Studies

Of the 42 studies included, 38 were peer-reviewed intervention studies, two were peer-reviewed review articles and two were theses (one for the degree of Doctor of Philosophy and one for the degree of Doctor of Audiology). Two of the intervention studies were published after the completion of their respective theses and each study is associated with and cited alongside its corresponding thesis [38,39,40,41]. Articles were published from 2008 to 2022. The country and design of each study are reported in Table 2.

### 3.3. Population Characteristics

Collectively, the age ranges of the participants spanned from 6 to 16 years old, but altogether, participants were aged between 7 and 12 years old for the majority of the studies (*n* = 27). Some included 6-year-olds [45,46,48,52,53], 13-year-olds [42,44,57,69,81], 14-year-olds [54,68,70], 15-year-olds [79] and 16-year-olds [43]. In most studies, children had NH acuity, with thresholds of 20 dB HL or better at 250 to 8000 Hz or from 500 to 4000 Hz. Children in eight of the intervention studies and both review studies had NH, but the criteria (decibels and frequencies measured) for NH were not described [42,43,44,54,63,67,71,72,76,79]. One study included 25 dB HL as the normal cut-off [78] and one study used a pure tone average (PTA) of less than 20 dB HL for 250 to 8000 Hz frequencies [40,41].

### 3.4. Identification of Auditory Processing Disorder or Difficulties

Participants identified as having an APD met the criteria by scoring two standard deviations below the mean on two AP tests or three standard deviations below the mean on a single test. It should be noted that the test batteries used varied among studies, and that the clinical guidelines followed were not always specified. In the studies that included participants with APD, eighteen studies focused on direct AT [38,39,43,49,50,51,57,58,64,66,68,69,70,71,77,78], three studies examined the use of a personal FM system or hearing aids [40,41,75,80] and three studies utilized multimodal interventions [42,44,76]. Reynold et al. (2016) conducted a review of studies focussing on children with APD; however, the review did not specify the criteria used to diagnose APD in the included studies [79]. Six of the studies focusing on AT and one focusing on the FM system required the presence of an APD, and also abnormal results at specific AP abilities targeted by the intervention: temporal patterning (Pitch Pattern Test [PPT]) and temporal resolution (Random Gap Detection Test [RGDT]) [49], SIN recognition or dichotic listening [50,51], SIN test (Speech-In-Noise in Indian English Test [SPIN-IE]) [57,58], dichotic listening (Staggerd Spondaic Words [SSW] Test) or the Test for APD in Children (SCAN) [77] and auditory figure-ground (AFG, subtest of the SCAN) or spatial processing abilities (specific subtests of Listening in Spatialized Noise–Sentence Test [LiSN-S]) [40,41].

The eligibility criteria for the remaining studies were based on the failure of one or more AP tests and not on an APD diagnosis. First, for the AT studies, some of the eligibility measures conducted were linked to the mechanism(s) addressed by the interventions. Inclusion criteria for AT studies involved obtaining abnormal results in tests assessing specific AP abilities, as follows:Dichotic listening (Dichotic Digits Test [DDT]) and temporal patterning (PPT) [52];Dichotic listening (DDT), temporal patterning (Pitch Pattern Sequence Test [PPST]), and auditory fusion (Auditory Fusion Test [AFT]) [53];Temporal resolution (RGDT), dichotic listening (Paediatric Speech Intelligibility—PSI test), and/or (Nonverbal Dichotic Test [NVDT]) [72];Dichotic listening only: DDT [74,81]; Persian Randomized Dichotic Digits Test (PRDDT), Persian Competing Words Test (PCWT), Persian Competing Sentences Test (PCST) [59]; and Competing Words Test (CWT), Randomized Dichotic Digits Test (RDDT) [62];DDT with abnormal right ear advantage (REA), temporal patterning (PPST) and selective attention (Monaural Selective Auditory Attention Test [mSAAT]) [65];Temporal patterning (Frequency Pattern Test [FPT]) [47];Temporal patterning (Persian PST [P-PST]) and dichotic listening (Persian-SSW [P-SSW]) [61];Spatial processing (LiSN-S) [45,46,48];Auditory figure ground (AFG test) [54];Dichotic listening, temporal patterning and selective memory (Multiple Auditory Processing Assessment [MAPA]), dichotic listening (Spectro-Temporal Modulation (STM) detection tasks) and SIN perception (Consonant–Vowel in Noise [CVN]) and Words in Noise Test [WIN]) [60];SIN perception (Monaural Speech Identification Test) [55];Temporal patterning (Duration Pattern Test [DPT]) [56].

Next, regarding a study focusing on a personal FM system [73], the eligibility criteria were based on a failure in one or two AP abilities: Dichotic listening (Double Dichotic Digits Test [DDDT]) or temporal patterning (PPST) [73]. Finally, for the study involving multimodal interventions [67], participants had to obtain results at one standard deviation below the mean on at least one of the following AP tests: Dichotic listening (Competing Sentences [CS]), SIN perception [monosyllables in noise], Temporal resolution [gap detection] and Binaural interaction (masking level difference [MLD]) [67].

Given that AP difficulties often coexist with other conditions, several studies listed the presence of concomitant disorders or challenges, including learning disabilities [52] or learning difficulties [70]; specific language impairment [69]; atypical phonological acquisition [72]; medicated ADHD, speech and language disorders and Chiari malformation [81]; difficulty with reading, apraxia, ocular and motor function difficulties, speech disorders, attention difficulties, or autism spectrum disorder (ASD) [49]; auditory memory, phonological, or language difficulties [38,39]; language impairment, reading disorders, autism, or ADHD [43]; Asperger syndrome, learning or language disabilities, or ADHD [75]; ADHD [77,80]; ASD, ADHD, or learning disabilities [79]; and language or reading disorders [42,44]. Children with ASD, Asperger syndrome or Chiari malformation were a minority in the specific studies that did not exclude these conditions. The above-mentioned conditions listed were not inclusion criteria. Twenty-two studies either reported or assessed normal or average intelligence levels, including measures such as the intelligence quotient (IQ), nonverbal intelligence, or psycho-intellectual abilities (AT: [38,39,46,47,52,53,55,56,60,64,65,66,69,70,81]; assistive devices and hearing aids: [40,41,75,80]; and multimodal interventions: [42,44,67]).

In Moossavi et al.’s (2015) study, which focused on improving working memory and SIN abilities, children who failed at the three subtests of the MAPA (DDT, PPST and mSAAT) and who had normal IQs were included, but children with neurological difficulties were not [63].

### 3.5. Assessment Tools for Measuring Outcomes of Interventions

The outcome measures utilized in the included studies to evaluate the effectiveness of the interventions are presented in Table A1 of Appendix B. The outcome measures are categorized into three main groups: behavioural outcome measures, neurophysiological outcome measures and questionnaires. Each type of outcome measure provides distinctive insights into various aspects of the participants’ responses to the intervention. In some studies, changes across training sessions were also monitored.
**Behavioural outcome measures.** Behavioural measures are divided into AP measures and cognitive, language-based and academic measures. AP measures are subdivided based on the ability assessed, while cognitive, language-based and learning measures are categorized by the type of skill evaluated, including language, academic, reading, attention and memory skills.

The majority of behavioural outcome measures that were identified were related to AP skills. In regard to AT targeting single and multiple abilities, each intervention studied had outcome measures related to the ability or abilities trained. Two AP test batteries and 28 AP tests were identified as outcome measures for AT interventions. Three studies used monaural low redundancy tests as outcome measures [69,70,71]. Some tests were available in different languages, such as English, French, Persian and Arabic. Temporal processing, specifically frequency ordering, emerged as the most frequently employed outcome measure, with eight different types of interventions utilizing this metric. Following closely after the frequency patterning tests, the DDT was the second-most-utilized measure, employed by seven types of interventions to assess their impacts. Frequency pattern tests measured significant improvements for different interventions, namely, musical training, SIN interventions, temporal patterning intervention, AT targeting multiple abilities, personal FM system, and multimodal interventions, but not for dichotic listening training. A significant improvement in DDT scores following dichotic listening interventions and AT targeting multiple abilities was observed, but no significant differences were measured following multimodal interventions and the use of a personal FM system.

A total of 21 outcome measures related to cognitive, language-based and learning abilities were identified. Language tests were used in the widest variety of intervention studies and the Clinical Evaluation of Language Fundamentals^®^—Fourth Edition (CELF-4) was the only outcome measurement tool assessing language that was used more than once.
**Neurophysiological outcome measures.** Eleven AEP protocols were identified to measure outcomes following interventions. Neurophysiological changes were assessed following ATs focusing on single or multiple AP abilities, the use of an FM system, or multimodal interventions. In general, after an intervention, an improvement in neural signal transmission is indicated by an increase in wave amplitude and a decrease in wave latency [42,82]. However, reduced amplitudes and earlier latencies observed in speech-evoked cortical responses could indicate a shift toward a more mature morphology following an intervention [58,83]. There is some evidence that AEP is an objective measure that can document the neurophysiological changes in the auditory system [42,49,82].

A study investigating the impact of singing lessons utilized auditory brainstem responses (ABR) associated with nonverbal stimuli (clicks) and complex sounds (syllables) [49]. Another used complex ABR (c-ABR), with a syllable being presented both with and without noise to evaluate the impact of AT focused on SIN, dichotic listening and temporal processing [69]. Waves I, III and V were identified and compared before and after training for click ABR, and waves V, A, C, D, E, F and O were examined for c-ABR. Significant changes were noted in both studies using c-ABR. Middle latency response (MLR), measuring components Na, Pa and the binaural interaction component (BIC), and long-latency AEP, such as P1-N1-P2-N2-P3 components, were also used as an outcome measure for lateralization training or AT targeting multiple abilities. Cortical AEPs, namely P1-N1-P2-N2, stood out as the most frequently used electrophysiological measure and demonstrated significant improvement following the majority of the interventions studied. Cortical AEPs were measured with clicks and syllables and with or without noise.
**Questionnaires.** Questionnaires are divided by respondent: teacher, parent, or child. Fifteen questionnaires were used as outcome measures, with parents and teachers being the most frequent respondents. Questionnaires answered by parents primarily focused on communication, auditory, and listening behaviours across various environments. Parents’ perceptions of changes in their child’s daily activities or listening abilities were assessed following various types of training, namely, SIN training, binaural processing training, AT targeting multiple abilities [70,72] and FM system use and fitting of binaural hearing aids. Significant changes were noted for binaural processing training and multiple-abilities training. One of the questionnaires, *Mesure des habitudes de vie* (MHAVIE; Translation: Assessment of life-habits) [50], assesses children’s lifestyle habits beyond listening and communication abilities by identifying specific activities of daily living that may pose challenges for them. Additionally, it helps determine whether the child requires help in performing these activities. The MHAVIE was utilized once for SIN training.

Teachers’ perceptions of changes in the children’s listening abilities in daily activities were also assessed to determine if various interventions, namely, binaural processing AT, SIN training, FM system use, and fitting of binaural hearing aids, had impacts on children’s auditory or listening skills. In one study [40,41], it was mentioned that the return rate of the questionnaires filled out by teachers was low, limiting the researchers’ ability to interpret or analyze the results effectively. Teachers noted significant changes following the use of the FM system, and the questionnaire used was the teacher version of the Listening Inventory for Education (LIFE) [75].

Two questionnaires assessed the children’s perceptions of change in listening abilities and daily activities, namely, the LIFE—Revised (LIFE-R), and the Speech, Spatial and Qualities of Hearing Scale questionnaire (SSQ). Improvements (nonsignificant for two studies [45,48] and significant for one [46]) were noted for binaural processing training. Significant improvements were observed by children after the use of a personal FM system [40,41,75].
**Performance over the training sessions.** Two studies evaluated the impact of binaural processing AT, while two others evaluated the effectiveness of SIN AT in terms of change across the training sessions. Improvement in performance during the training was significant [45,46,51] and non-significant [50], according to the authors.**Statistical significance of the results.** The statistical significance of the results at the different outcome measures listed in the studies is reported in Table A1. Results following the interventions were significant (S), non-significant (NS), or not different (ND) (indicating no discernable impact of the training on these measures) for the experimental group. One outcome measure (CNA) could not be analyzed because of the low return rate of questionnaires. For the sake of conciseness, the rating S or NS was assigned when the majority of subtests within a test battery measured S or NS changes, even if some subtests were ND. Conversely, the rating NS was assigned in the opposite situation. For neurophysiological tests, if changes were noted (regardless of whether the change was for latency or amplitude, with certain stimuli only, or in one or both ears), S or NS was assigned according to the results. Readers are directed to consult both the table of study characteristics and the individual studies for comprehensive details on the degree of change.

### 3.6. Intervention Characteristics

Interventions listed in this scoping review can be grouped into six categories: (1) single-ability auditory training, (2) multiple-ability auditory training, (3) assistive devices (FM system), (4) hearing aids, (5) top-down interventions and (6) multimodal interventions. Table A2 of Appendix C describes each training or intervention and schedule in greater detail.

#### 3.6.1. Single-Ability Auditory Training

Seven studies examined the impact of dichotic listening training [59,62,65,74,77,78,81]; two focused on localization and lateralization abilities [64,66]; and three investigated the impact of spatial processing intervention [45,46,48]. One examined the effect of singing lessons on pitch and rhythm development [49], one evaluated musical pitch training [47], one focused on temporal patterning training [56], one focused on phonemic training [61], and nine evaluated the impact of SIN training [38,39,50,51,54,55,57,58,60,71].
**Dichotic listening training.** The interventions aiming at improving dichotic listening used two approaches. The first one involved presenting different stimuli to both ears at various interaural time intervals [59,65,74,77,78]. As the training progresses and the child successfully completes tasks, the stimuli are gradually presented with decreased time intervals until they are eventually presented simultaneously. In some training programs, the stimuli start in the better ear [59], while in other programs, there is a variation in which the ear that receives the stimuli first (leading ear) may change [74]. Participants are asked to repeat what they hear in both ears (binaural integration tasks) or in one ear only (binaural separation tasks). Stimuli and presentation media vary from one program to another. Children completing interaural timing difference training have dichotic deficits, which are demonstrated by deficits in at least one ear at the DDT [59,74] or SSW and/or SCAN [77,78], or abnormal REA scores in the DDT [65].

In Barker and Bellis (2018), children who initially had dichotic deficits, as evidenced by abnormal DDT scores, demonstrated significant binaural improvements in the DDT, and most notably in the left ear, after undergoing dichotic interaural timing difference training [74]. A similar scenario was observed in Mahdavi et al. (2021) with children who had dichotic listening deficits, as noted by one abnormal result in at least one Persian-adapted dichotic test (PRDDT, PCWT and PCST) [59]. Following dichotic interaural timing difference training, the scores for these dichotic tests were significantly better bilaterally post-intervention, especially for the non-dominant ear [59].

In Shoemaker’s (2010) thesis, children who had dichotic deficits, according to abnormal results for the SSW and/or the SCAN, demonstrated significantly better scores in the SSW and a dichotic AT screening tool for both ears when compared to a control group following dichotic interaural timing difference training [77]. They also demonstrated significant improvements in scores for competing words and filtered word tests, outperforming the control group.

In Stephenson’s thesis (2008), children identified with an APD through a test battery that included the SSW and the SCAN underwent dichotic interaural timing difference training. Children showed significant improvement only in the left competitive condition on the SSW [78]. Additionally, scores on a dichotic AT screening tool significantly improved, post-training [78]. Results on the SCAN post-training were not significantly different from the pre-training assessment.

The impact of dichotic offset training, another version of dichotic interaural timing difference training, was studied by Delphi et al. (2018) in children who had abnormal REA scores, DDT scores, PPST scores and mSAAT scores. The results suggest that this training was effective in improving left-ear performance in the DDT and in decreasing REA [65]. Dichotic offset training was compared with another method aimed at enhancing dichotic listening through varying interaural intensities, namely, dichotic interaural intensity difference training. The results suggested that dichotic offset training required more sessions than dichotic interaural intensity difference training to achieve an equivalent level of improvement [65].

Some studies found that the specific skills trained through dichotic interaural timing difference training can generalize, indicating improvement in daily life functioning and improvement in skills that were not directly targeted by the training. Indeed, in Barker and Bellis’s (2018) study, anecdotal comments by parents suggest that children had more confidence in the classroom and followed complex directives better at home [74]. Improved recognition of filtered words, as measured by the SCAN, was also noted post-intervention in one study [77], but not in another [78].

The second approach aiming to improve dichotic listening involved interaural intensity differences [62,65,81]. The intervention involves presenting different stimuli simultaneously to both ears, with varied interaural intensities, to encourage the use of the weaker ear. As the child achieves success in completing tasks, the intensity of the stimuli in the better ear increases progressively. Participants are asked to repeat what they hear, either in both ears, or in one ear only. Stimuli and presentation media vary from one program to another. In the studies utilizing the dichotic interaural intensity approach, children completing the intervention had abnormal REA scores on the DDT [65], significant interaural asymmetry at the DDT (left ear 10 to 20% poorer than right ear, according to the age of the participants) [81], or 25% binaural asymmetry in the Competing Words Test (CWT) and the RDDT (dichotic tests) [62].

In Delphi et al. (2018), the impact of dichotic interaural intensity training in children who had abnormal REA, DDT, PPST and mSAAT scores was investigated. This training method was effective in enhancing scores in the left ear on the DDT and in reducing REA [65]. In comparison to dichotic offset training, this method required fewer training sessions to attain a comparable level of improvement [65].

Moncrieff and Wertz (2008) also studied the impact of dichotic interaural intensity training in children who had unilateral dichotic deficits with significant REA at the DDT. Following the intervention, children had improved dichotic listening skills in both ears and scores on the DDT and CW were significantly better in the left ear, resulting in symmetrical performance between ears post-training [81]. Improvements were also maintained one year post-training [81]. Moncrieff and Wertz (2008) found benefits associated with an intervention on language skills, phonological awareness and filtered word recognition [81].

Furthermore, Nazeri et al. (2020) examined the effects of dichotic interaural intensity difference training on a group of children diagnosed with amblyaudia. This condition was characterized by abnormal and asymmetric results in dichotic tests, namely, in the case of this study, the CWT and RDDT. Participants had 25% interhemispheric asymmetry in these measured parameters. Following the AT, significant improvements were observed in the SSW, as evidenced by a reduction in errors post-training compared to pre-training performance measures [62].
**Localization and lateralization training.** Two studies examined the impact of localization and lateralization training on spatial processing, localization and lateralization abilities [64,66]. The studies involved the same group of participants and the same intervention. The training focused on detecting and identifying sounds presented under headphones with various interaural time differences, which created the perception that the sounds were coming from distinct locations. To be included in the studies, results for the DDT (dichotic listening), PPS (temporal patterning) and mSAAT (auditory attention) had to be two standard deviations below the mean [64,66]. Following the intervention, the experimental group demonstrated significant improvements in auditory memory scores bilaterally [64,66], and in spatial word recognition in noisy conditions, when compared to the control group [64]. Noticeable neurophysiological changes were observed in the experimental group, compared to the control group. These changes were evidenced by a significant reduction in latency and an increase in amplitude, as measured by the BIC [66]. In summary, the benefits of the training were noted for binaural processing, SIN abilities and auditory memory.**Spatial or binaural processing training.** Three studies examined the efficacy of the Listening in Spatialized Noise Training (LiSN & Learn) [45,46,48]. This AT program is designed to enhance binaural processing through SIN activities, with target stimuli presented from the front, and the noise from various directions. This program aims to improve elements of binaural processing, such as interaural timing and intensity differences. These elements are crucial for spatial processing, allowing individuals to attend to stimuli originating from one direction while ignoring sounds or noises from other locations [46]. Participants had spatial processing disorder (SPD), diagnosed with the LiSN-S [45,46,48]. SPD is a subtype of APD and is “characterized as a reduced ability to utilize cues important for accurate localization and listening in noise” (p. 376, [46]).

In Cameron and Dillon (2011), children who had SPD completed the Listen & Learn training program. Following the intervention, a significant improvement in the LiSN-S in the conditions where speech is spatially separated from the distractors was noted [46]. Improvement was maintained three months post-intervention. Improvements were nonsignificant in attention abilities but significant in memory [46]. In addition, the children who underwent the training demonstrated significant improvement in their ability to understand speech, both in quiet and in noise, as reflected by their responses to the SSQ [46].

In another study, a group of children with SPD completed the Listen & Learn training program, while another group of children with SPD completed the Earobics Home Version AT software (Step 1) [84]. The experimental group who completed the Listen & Learn training program had significant improvements in the LiSN-S in the conditions in which speech is spatially separated from the distractors, compared to the control group, who completed another AT [45]. No improvement was noted for either group in conditions in which speech and distractors came from the same direction [45]. Children and teachers filled out the LIFE-R questionnaire. For both groups, improvements in listening skills were noted by the children and the teachers. However, more improvements were noted in the experimental group [45]. The parents of the children in the experimental group perceived significant improvements in AP behaviours post-training according to the Fisher’s Auditory Problem Checklist (FAPC) [45].

In Graydon et al.’s (2018) research, children with SPD who completed the Listen & Learn training program had significant improvements in the LiSN-S in the conditions in which speech is spatially separated from the distractors, compared to a control group who did not complete an AT [48]. Parents, children and teachers’ perceptions of changes in daily listening activities were measured with the FAPC, the LIFE-R, and the Teacher Evaluation of Auditory Performance (TEAP). All noticed improvements in daily listening skills, but these were only significant according to data from the parents [48].

Overall, the benefits of the LiSN & Learn training were noted for spatial processing abilities and for higher-order abilities, such as attention and memory. Parents, children and teachers also perceived improvement in daily life-activities [45,46,48].
**Musical training.** The impact of singing lessons on the subcortical auditory response to the clicks and speech sounds of children who have an APD and, more precisely, difficulties with temporal processing, was investigated by Koravand et al. (2019) [49]. While no significant improvement was observed with click ABR post-training, the magnitude of various subcortical responses measured with speech stimuli showed improvement in the majority of the participants after training, suggesting the benefits of singing lessons for the central AP [49].

Tomlin and Vandali (2019) investigated the effectiveness of musical pitch training to improve temporal patterning abilities in children with temporal patterning deficits. Participants completed a modified version of the aTune musical-pitch training program, an AT during which children engage in activities involving the discrimination and identification of pitch, timbre and instruments [47,85]. Significant improvements in temporal patterning were noted post-training and were maintained 60 weeks after the training. No significant changes were noted in reading fluency, attention, or memory abilities, or for children’s perception of listening difficulties according to their LIFE-R scores [47].
**Temporal patterning training.** In their research, Maggu and Yathiraj (2010) aimed to improve temporal patterning abilities with temporal patterning training. The participants were children with low scores on the DPT [56]. The training activities involved discriminating and identifying tones based on variations in both frequency and duration [56]. Compared to a control group, the children who completed the training demonstrated significant improvements in the DPT, suggesting an amelioration of temporal patterning abilities [56]. The results in the other AP measures (auditory integration and separation, and gap detection) did not change post-training, but improvements were noted in auditory memory skills, suggesting that benefits generalized to other skills [56]. Improvements were maintained one month following training [56].**Phonemic training.** Negin et al. (2018) studied the effects of phonemic training on a child experiencing difficulties in phonemic awareness and decoding, as assessed by the P-PST and SSW. In this single-subject design study, a child learned various phonemes presented on their own, and eventually with other phonemes and in words. Different strategies were presented to the child to learn the phonemes. Positive effects of the training were observed in binaural integration abilities [61]. However, there was no improvement in phonemic awareness abilities [61]. Nevertheless, a significant reduction in the number of phoneme errors was noted [61]. Additionally, there were reported improvements in academic performance, particularly in spelling skills, following the training [61]. These improvements were sustained two months after the intervention [61].**Speech in noise training.** Nine identified studies investigated the impact of SIN training on noise tolerance and speech-recognition-in-noise abilities of children with AP difficulties or an APD, specifically those struggling with listening in noisy environments.

In their exploratory study, Jutras et al. (2019) examined the impact of a SIN training software (http://monleb.com/) called *Logiciel d‘écoute dans le bruit* (LEB) [86] on children’s SIN performance, electrophysiological measures and daily listening activities, according to questionnaires filled out by teachers. This listening-in-noise training program is computer-based and can be completed in a clinical setting, at home, or at school. The activities include word or sentence identification as well as understanding directions and short stories, and are completed in the presence of a babble noise [51]. Children with an APD and difficulty with SIN or dichotic listening completed the LEB at school. Children completing the training exhibited significant improvement in noise tolerance across sessions, but no significant improvement was observed with the Hearing in Noise Test (HINT), a measure of SIN perception. However, more improvement in the HINT for the training group, compared to the control group, was observed. Additionally, 40% of children had a noticeable increase (0.5 µV) in P1 and N2 amplitudes post-training, compared to 17% for the control group. Following training, P1 and N2 latencies decreased by at least 10 ms for 40 to 50% of the children, compared to 17% in the control group. These results suggest potential neurophysiological changes for the experimental group, but the changes were not significant. Teachers did not notice significant changes following the training according to their answers in two questionnaires, namely the Scale of Auditory Behaviours (SAB) and the Screening Instrument for Targeting Educational Risk (SIFTER). Nonetheless, individual data indicated that the experimental group showed greater improvement than the control group in their abilities to discriminate, identify and understand rapid or muffled speech, as observed through the questionnaires [51].

Brasil and Schochat (2018) studied the effectiveness of the Programa de Escuta no Ruído (PER), a translated and adapted version of the LEB for Brazilian Portuguese speakers, with children who have APD [87]. Children completed two pre-training assessments for control measures. After the intervention, significant improvements were noted in all AP measures, namely SIN, PSI, SSW and FPT. Improvements were noted in academic skills post-intervention, as noted by the School Achievement Test (SAT), but were not significant [71].

Loo et al. (2012, 2016) also studied a computer-based AT program focusing on enhancing SIN abilities in children with an APD [38,39]. This at-home training program included speech activities in the presence of competing sounds. Compared to the control group, the experimental group showed significant improvements in speech reception thresholds (SRTs) in the LiSN-S. Additionally, the experimental group’s mean SRTs slightly improved three months following the intervention. After the training, there were significant improvements in parents’ perception of their children’s pragmatic abilities, as indicated by their responses in the Pragmatic Profile questionnaire. Additionally, teachers reported significant improvement post-training for the experimental group in classroom listening abilities, as reflected in their responses to the Children’s Auditory Performance Scale (CHAPS) [38,39].

A pilot study by Jutras et al. (2015) examined the impact of a SIN AT that was completed in a clinical setting and consisted of auditory activities, such as syllable or word discrimination, word identification, memorization and understanding directions. Activities were completed with various noises delivered through speakers, utilizing an audiometer and a CD player [50]. Children participating in this pilot study had an APD and difficulty with SIN or dichotic listening. Following the training, the results suggested that children became more tolerant to noise levels as the therapy progressed because the percentage of correct responses improved, and the noise levels increased. After the intervention, no change was noted for the scores on the HINT, compared to the control group. Measures of cortical AEP suggested a shorter P1 latency, and an increased N2 amplitude post-training for the experimental group, compared to the control group. These results suggest that neurophysiological changes were noted following the SIN AT and that the training improved neural transmission of auditory stimuli. Six months following the intervention, P1 latency remained stable and N2 amplitude increased. After the intervention, there was no observed change in auditory and social abilities, as reported by both the parents and teachers in the French adaptation of the SAB and the MHAVIE. Additionally, teachers did not report any change with the SIFTER [50].

Hassaan and Ibraheem (2016) investigated a semi-formal AT with children presenting auditory figure-ground deficits [54]. During the training, the children listened to a story presented by the examiner at the same time as a competing noise and answered questions afterward. Post-training measures showed significant improvement in SIN measures, competing speech tests and temporal patterning abilities. Speech-evoked neurophysiological measures suggested improved neural transmission of auditory stimuli. A significant increase in the peak-to-peak amplitude was observed, as well as a decrease in the thresholds in noise for the P1-N1 complex [54].

In Afshari et al. (2022), the effectiveness of a SIN AT utilizing auditory spectro-temporal modulation (STM) identification games was examined [60]. The participants were children who had abnormal performance in three MAPA subtests: DDT, PPS and mSAAT. Additionally, these children displayed abnormal STM detection thresholds and difficulty in Persian versions of SIN tests, namely, CVN and WIN. Following training, STM detection thresholds improved significantly compared to the control group, which did not follow any therapy, and changes remained stable one month post-intervention [60]. These results suggest improvement in the detection of spectro-temporal modulation abilities. The results of SIN testing (CVN and WIN) showed significant improvement for the experimental group, compared to the control group, and only CVN scores remained stable one month post-training [60].

Two studies from Kumar et al. (2021) examined the effectiveness of an interactive computer-based noise desensitization training [57,58]. During this training, children listened to lists of words at various SNRs and in competing noises. Six images were displayed, and children were asked to identify words in noise by selecting a corresponding image. In both studies, children had an APD and abnormal scores on the SPIN-IE [57,58]. In the first study [57], the results indicated a significant improvement in the SPIN-IE, Gap Detection Test (GDT) and DPT measures for the experimental group, compared to the control group. In addition, no change was observed in the Revised Auditory Memory and Sequencing Test (RAMST) and the Dichotic Consonant Vowel (DCV) measures. The impact of the training on digit span (auditory memory) was also examined and revealed significant improvements for the experimental group, except for forward digit span. In summary, the SIN training significantly improved temporal auditory skills and SIN skills [57]. In their second study [58], the results showed significant improvements in SPIN-IE, GDT and DPT measures for the experimental group compared to the control group. However, no change was observed for RAMST (auditory memory and sequencing) and DCV measures. Post-intervention neurophysiological changes were reported for the experimental group, but not for the control group, when measured with speech stimuli (syllable /da/). Compared to pre-intervention measures, there were significant decreases in P1 and N2 amplitudes in the quiet condition, and in all four peak (P1-N1-P2-N2) amplitudes in the noise condition. The authors suggest that these changes in amplitude follow a pattern of maturation of the cortical responses to speech stimuli. Subjective feedback was provided by both parents and teachers. While parents did not observe improvement in their children’s listening skills after training, both participants and teachers reported positive changes. Participants noted reduced stress, and teachers observed increased responsiveness to instructions. Hence, improvements gained during training also transferred to daily activities [58].

In Maggu and Yathiraj (2011), children who had difficulty identifying monosyllabic words in noise listened to recorded texts presented with various types of noises and at various SNRs [55]. The authors did not specify whether the training was performed under headphones or with speakers. Questions were posed to the children after the text presentations. Following the training, the experimental group had significantly increased their scores in SIN tasks, including monaural monosyllabic words in noise, sentence identification in noise, and sound field speech identification. No change was noted for the control group, which did not undergo any training [55].

#### 3.6.2. Multiple-Ability Auditory Training

Six intervention studies examining the effects of AT programs designed to target various AP abilities were identified in this scoping review. Two studies investigated the outcomes of AT programs with the goal of improving temporal processing and phonemic awareness abilities [52,53]. The participants in the first study were children experiencing difficulties in dichotic listening and temporal sequencing, as shown by abnormal results on the DDT and PPT [52]. The children were divided into two experimental groups, each undergoing different intervention modalities. One group completed a computerized training of auditory temporal processing and phonemic awareness, while the other had phonemic awareness and auditory-directives training with a therapist. Post-intervention, both groups demonstrated significant improvements in PPT and DDT scores and phonemic awareness abilities, as well as a decreased P1 latency [52]. The improvements in the scores on the DDT suggest that the benefits of both interventions were extended to binaural integration abilities [52]. In the second study, participants were children with an APD and were divided into two experimental groups, each following a different intervention modality. One group underwent computerized training in auditory temporal processing and phonemic awareness, while the other group received auditory temporal processing and phonemic awareness training with a therapist [53]. Statistically significant improvements post-intervention were noted for both groups for AP measures, including the DDT, PPST and AFG (except 250 Hz); phonemic awareness abilities; a decreased P1 latency; and improved AP perception as evidenced by an APD questionnaire [53]. Benefits from the interventions were extended to binaural integration and auditory fusion abilities. No significant differences in the outcome measures were observed between the two groups post-intervention [53].

Two other studies involved AT programs aiming at improving dichotic listening, temporal processing, auditory figure-ground and SIN abilities [68,69]. In Donadon et al. (2019), children who had an APD underwent computerized AT activities. These activities focused on enhancing binaural integration, temporal resolution, and ordering, as well as figure-ground processing [68]. The control group underwent visual training. The AT program administered to the experimental group yielded statistically significant improvements in various AP abilities [68]. Specifically, the DDT, FPT, Gaps-in-Noise (GIN) Test and Synthetic Sentence Identification with Ipsilateral Competitive Message (SSI-ICM) measures demonstrated significant improvements in the experimental group, suggesting that the AT program produced changes in AP abilities [68]. Filippini et al. (2012) investigated an AT program that included competitive speech perception and SIN tasks, dichotic listening exercises and temporal processing involving frequency and duration aspects [69]. The AT was completed with an audiologist, using an audiometer [69]. Children who completed the intervention were divided into two groups: APD with specific language impairment (SLIa group) and APD without specific language impairment (APD group). The control groups included one with neurotypical children and one with an APD and a specific language impairment (SLIb group). The SLIb group underwent speech therapy. AP abilities of children in both of the experimental groups improved significantly after the training. Post-intervention scores in the SIN test, SSW, DDT and PPST were significantly better for both groups, compared to the control groups. The c-ABR measures were completed, with a syllable presented both in quiet and in noise, before and after the intervention. Compared to pre-intervention measures, both experimental groups had decreased latencies when ABRs were measured with syllables in noise; specifically, the APD group had earlier V and D latencies and the SLIa group had earlier V, C, D and E latencies [69]. There were no significant changes in latencies observed for the measures when the syllable was presented without noise. The only exception was the latency E in the SLIa group, for which the latency was earlier post-intervention. The authors concluded that the program was shown to be beneficial for children with an APD, as it improved AP skills and neural transmission of auditory stimuli.

Training programs involving dichotic listening, temporal processing, auditory figure-ground/SIN training and localization were studied by Schochat et al. (2010) and De Melo et al. (2018). In Schochat et al. (2010), the experimental group, consisting of children with an APD, was compared to a control group of children without an APD and who completed an informal training with parents at home. Post-intervention, children in the experimental group had increased scores on AP measures, including PSI, SIN tests, the DDT, SSW and NVDT [70]. They did not show neurophysiological changes in terms of MLR latencies measured with clicks, but there was a significant increase in Na-Pa amplitude, suggesting that the AT had an impact on auditory neural processing [70].

In De Melo et al. (2018), children with an APD underwent a computerized training program. Children were divided into two groups: APD with typical phonological acquisition, and APD with atypical phonological acquisition. Neurophysiological changes were noted following training for both groups [72]. Children who had a typical phonological acquisition had a significant decrease in N2 and P3 latencies in the left ear, and children with atypical phonological acquisition had a significant decrease in P2 latency in the right ear following therapy [72]. P3 latency decreased for both groups following training, but only significantly for the left ear in children with typical phonological acquisition [72]. Some children had no P3 pre-training, but a measurable one post-training [72]. The parents’ responses on the SAB questionnaire indicated improvements in functional auditory behaviours after the intervention [72].

#### 3.6.3. Assistive Devices

In this scoping review, three intervention studies [40,41,73,75] and one review study [79] examining the use of assistive devices to improve speech perception in the classroom in children with an APD were identified. In all studies, subjects used a personal FM system. The fitting of the system consisted of a microphone worn by the teacher and receivers worn by the children.

In Umat et al. (2011), children with AP difficulties used an FM system for 12 weeks and were tested pre-fitting, 12 weeks post-fitting and 1 year after not using the FM system. Half of the children utilized monaural fitting and the other half utilized binaural fitting. Compared to the control group, which consisted of children with AP difficulties who did not use an FM system, the benefits of wearing an FM system were noted for both experimental groups for working memory (Rey Auditory Verbal Learning Test, RAVLT) after 12 weeks and in the long-term (1 year after post-fitting measures). No significant difference was noted between the two experimental groups at twelve weeks post-fitting, but improvements in working memory abilities increased for the monaural fitting group 1 year following the use of the FM system. In the RAVLT, long-term benefits were noted in two subtests: best learning and retention of information. The authors concluded that the use of an FM system can improve short-term auditory memory abilities of children with a suspected APD [73].

Smart et al. (2018) investigated the impact of using an FM system in the classroom on children with an APD. Participants wore binaural open-fit receivers. During each assessment, children completed the measures with and without the FM system. Assessments were completed twice before the fitting for baseline-control measures, and once five months after the fitting. When children were wearing the FM system during testing, immediate benefits were observed with speech in spatial noise (SSN) easy words and the FPT, but not with MLD, GIN, DDT, compressed and reverberated words (CRW), SSN hard words or the Integrated Visual and Auditory Continuous Performance Test (IVA-CPT). After wearing the FM system for five months, no significant changes were noted on the following tests: MLD, GIN, DDT, CRW, SSN hard words (also known as infrequently heard words) and sustained attention score (IVA-CPT). Additionally, there were no significant changes with respect to cortical AEP in P1-N1-P2-N2 amplitudes and latencies. However, significant improvements were noted in temporal processing (FPT) and SSN easy words. When measuring AEP with a syllable in noise and without the use of the FM system, P1 and N2 latencies were longer, and N2 amplitude decreased compared to the measurements taken with the syllable in the absence of noise. When the children wore the FM system, P1 and N2 latencies and N2 amplitude were less affected by the noise. According to their responses on the LIFE-UK, significant improvements were observed in the children’s and teachers’ ratings of classroom listening abilities following the use of the FM system. Parents also reported overall improvement in students’ listening after five months of wearing the FM system [75].

Stavrinos (2019) completed two intervention studies. In both, the use of binaural open-fit remote microphone hearing aids was explored with children who had AP difficulties and SIN and/or spatial processing difficulties. In the first study, both children and their teachers used the system at school, five days per week, over a span of three months. Compared to the control group, consisting of children with an APD who did not use the FM system, no significant change was noted for SIN abilities (AFG), attention abilities (Test of Everyday Attention in Children, TEACh) and memory abilities (Automated Working Memory Assessment, AWMA) for the experimental group after three months. Parents filled out questionnaires, namely, the Children’s Communication Checklist—Second Edition (CCC-2), and the Children’s Auditory Performance Scale (CHAPS). Their responses suggest that they did not notice significant changes in listening or language abilities following intervention, but they did notice some improvements in attention at home. Children subjectively reported some improvement in their daily listening situations at school, according to their answers on the LIFE-R [40,41].

In Stavrinos’s (2019) second study, children also had AP difficulties and SIN and/or spatial processing difficulties and used binaural open-fit remote microphone hearing aids. Outcome measures were assessed before the fitting, three months post-fitting and six months post-fitting for both the experimental group, which used the FM system, and the control group, consisting of children with APD who did not use an FM system. Compared to the control group, the experimental group did not show improvement in SIN or spatial listening conditions, as evidenced by the LiSN-S, or in attention abilities (TEACh) or parent’s perception of listening and language abilities, as per their answers on the CHAPS and CCC-2. As indicated by the LIFE-R, children in the experimental group noted significant improvements in classroom listening situations at three and six months post-fitting [40,41].

Finally, in Reynolds et al.’s (2016) review, moderate evidence supports the use of a personal FM system for children with AP difficulties. Based on seven studies, the review indicates that the FM system may improve academic outcomes, especially those related to speech perception and recognition in the classroom and classroom listening behaviours. Only two studies in Reynolds et al. (2016) met the inclusion criteria of the present scoping review. They were conducted by Umat et al. (2011) and Sharma et al. (2012), and are discussed in the review [79].

#### 3.6.4. Hearing Aids

Kuk et al. (2008) examined the impact of the wearing of binaural mild gain open-fit hearing aids by children who have an APD and NH thresholds [80]. Half of the children were wearing the hearing aids consistently at school. SIN tasks included NU-6 word lists in speech-shaped noise presented with loudspeakers and various SNRs. Measures were completed with three different microphone settings: omnidirectional, omnidirectional with noise-reduction algorithm and directional with noise-reduction algorithm. Amplification and omnidirectional microphones did not improve SIN abilities, but performance in SIN tests was significantly better when directional microphones were used and the noise reduction algorithm was activated. Performance in SIN tasks did not improve over the following months. Over time, a slight improvement was noted in attention abilities in noise, as noted when the Auditory Continuous Performance Test (ACPT) was performed with noise presented behind the participants. This slight improvement was observed when the measures were completed with directional microphones and the noise reduction algorithm activated. Teachers and parents completed the CHAPS. Teachers noted improvements in quiet and noisy situations, and parents noted improvements in memory and attention abilities. The authors concluded that the use of hearing aids with the noise-reduction algorithm and directional microphones improved SIN abilities [80].

#### 3.6.5. Top-Down Intervention

Moossavi et al. (2015) studied the effectiveness of a working memory training program in improving SIN abilities and auditory working memory abilities [63]. The participants included children with difficulties in dichotic listening (DDT), temporal patterning (PPST) and auditory attention (mSAAT). Following the training, the experimental group, compared to the control group, which did not complete an intervention, had significant improvements in auditory stream segregation abilities (concurrent minimum audible angle, CMAA) and significant improvements in working memory abilities (a Persian non-word repetition test and digit span subtests of the Wechsler Intelligence Scale for Children) [63].

#### 3.6.6. Multimodal Approaches

Four intervention studies [42,44,67,76] and one review study [43] examining multimodal approaches were identified. In Bellis and Anzalone’s (2008) case study, a child with an APD followed a multimodal program aimed at improving AP abilities, top-down abilities and the listening environment [76]. Modifications in the classroom, such as preferential seating, and strategies for the teacher, including the use of visual cues and the pre-teaching of new vocabulary, as well as a personal FM system in the classroom, were recommended. Parents and teachers used clear speech when interacting with the child. The computerized auditory and language training program Earobics [88] was completed. Listening and problem-solving skills were trained. Following the program, improvements were noted in auditory closure abilities (low-pass filtered speech, LPFS), as well as dichotic listening abilities, including binaural separation (CST) and integration (DDT) [76].

Two studies examined the impact of a multimodal approach aiming at improving discrimination, temporal processing, phonemic awareness, language abilities and the listening environment. In the first study, children with an APD were included and were divided into four experimental groups: discrimination training, discrimination training and FM system, language training, and language training and FM system [44]. A control group, consisting of children with an APD who did not follow any intervention, was included. Post-intervention, no differences were noted in the measurements of the HINT and Comprehensive Assessment of Spoken Language (CASL) for any group [44]. Significant improvements were observed for both discrimination and language training, and the use of the FM system provided an additional advantage. Benefits were identified for language, AP (FPT), and phonological skills domains. In both discrimination groups, a significant improvement in core language in the CELF-4 was noted. Similarly, in both language groups, significant improvements were observed in sentence recall (CELF-4) and nonword spelling (Quick Interactive Language Screener, QUIL). Parents and participants subjectively reported students’ increased interest in reading and greater responsiveness and success in class [44].

Sharma et al. (2014) investigated the impacts of these interventions on cortical AEP. The same five groups from the previous study completed neurophysiological measurements with a syllable heard both in quiet and in noise. These measures were completed twice before the intervention for baseline control measures. P1 and N250 latencies were stable, but N250 amplitude significantly decreased at the second assessment in both quiet and noisy conditions. Sharma et al. (2014) reported a significant change in N250 amplitude across the three measures for children who completed the interventions, and the amount of change was associated with the different forms of training. Discrimination training might have a greater impact on the amplitude of cortical AEP, compared to top-down language training, according to the authors [42]. N250 amplitude was significantly smaller post-intervention for the discrimination group when measured in quiet and in noise [42].

In Putter-Katz et al. (2008), two groups of children completed an intervention focusing on both top-down and bottom-up approaches [67]. Bottom-up approaches included AT for SIN and attention, while top-down ones included coping strategies, accommodations at home, and speech reading. Additionally, an FM system was used (the authors did not specify where it was used). Children were divided based on their listening difficulties: one group had SIN difficulties and the other group had SIN and dichotic listening difficulties. The control group experienced difficulty with SIN, but not with dichotic tasks. Following the intervention, children from both groups showed significant improvement in SIN tasks. The SIN-and-dichotic-listening-difficulty group significantly improved in dichotic competing tasks, whereas the SIN-difficulty-only group showed some improvement. No change was noted for the control group [67].

Out of the seven studies included in Wilson et al.’s (2013) systematic review on neurophysiological outcomes of AT or language training, only one met the inclusion criteria of the current scoping review [43]. This study featured a multimodal intervention focusing on dichotic listening, temporal processing, SIN, binaural processing and language abilities. The latency of P300 evoked by tone-bursts improved post-intervention and may be sensitive to behavioural changes in AP. Improvements were noted in AP tests, namely SSI-ICM, SIN, SSW and NVDA [43].

## 4. Discussion

This scoping review aimed to systematically map, compile and synthesize recent published research on interventions for school-aged children with NH thresholds and APD or AP difficulties. In total, 42 studies were listed, including 38 peer-reviewed intervention studies, two review articles and two theses. In most, an experimental group was compared to a control group either completing a placebo intervention or receiving no intervention at all. Only eleven studies conducted long-term follow-up assessments.

In 24 studies, the presence of an APD was an inclusion criterion. However, the protocols and the psychometric tests used to determine the presence of an APD varied across the studies, which may be consistent with the lack of consensus on the appropriate test battery, as discussed in the Introduction. (See Appendix A for details on the metrics used to diagnose APD.) When having an APD was not an inclusion criterion, difficulties in one or more AP skills were present for all participants. The most commonly reported diagnoses concomitant to AP disorder or difficulties were speech or language problems, ADHD and learning disabilities. Some studies did not clearly indicate whether the children had one or more concomitant diagnoses, while others did not specify their exclusion criteria in regard to concomitant conditions. The procedure for diagnosing APD, as well as the presence or absence of concomitant disorders, will influence the inclusion and exclusion criteria, thus shaping the outcomes following the intervention studied. Employing a standard battery of tests and differentiating APD from other concomitant problems would provide a deeper understanding of how interventions directly impact AP, and guide the tailoring of appropriate intervention strategies [10]. However, listening difficulties in children with APD are often multifaceted, and may coexist with other diagnoses, underscoring the importance of considering these complexities when aiming to improve the individual’s participation in daily life [8,10]. Certain tools, such as various electrophysiological measures, appear promising in identifying APD when combined with psychometric tests [89]. Nevertheless, further studies are needed to determine the electrophysiological characteristics of children with APD [89] or broader listening difficulties [90]. Additionally, it remains to be determined whether deficits in electrophysiological measures correlate with the level of listening difficulties reported by children in daily life. Gaining a better understanding of the nature of listening difficulties in children with APD would increase the internal validity of intervention studies and ensure the generalizability of the findings to the target population.

The majority of the identified interventions aimed to rehabilitate specific or multiple AP abilities through a bottom-up approach, namely AT. The most widely researched AT was SIN training, and findings suggest that it can be conducted with various stimuli, noises and activities. AT was primarily delivered through a computerized format, but it could also involve a therapist. Findings indicate that AT interventions are significant, practical and generally effective. However, they only partially address the listening challenges experienced by children and their families. While these interventions show an impact on the targeted skills, it remains unclear whether the improvements measured can be transferred to other skills, such as different AP abilities or cognitive and language skills.

For FM systems, significant improvements were observed in AP and memory abilities. The teachers’ and the children’s perception of listening abilities also improved. No improvement was observed with AEP when measured without the device, however, indicating that the FM system did not prove beneficial at improving the neural processing of auditory information.

Hearing aids were also investigated. Significant improvements in SIN abilities were observed when directional microphones and noise-reduction algorithm were activated. However, these improvements did not change over time. Some improvement was noted in auditory attention in noise over the course of the study. Parents and teachers also reported improvements in the children’s listening abilities across various listening situations.

As previously discussed, efficient listening is a skill that requires the orchestration of both bottom-up and top-down processes. Most interventions that are studied primarily involve bottom-up approaches or utilize devices that enhance access to essential or pertinent sounds. Although these studies also examined the impacts of such interventions on higher-order processes such as language, memory, attention, reading and academic skills, the results are generally inconclusive.

A limited number of top-down interventions were identified. Specifically, six studies, five of which investigated multimodal interventions, explored the impact of targeting higher-order processes, including language, memory and problem-solving abilities, to mitigate the challenges associated with an AP disorder or difficulties. As shown in the multimodal interventions that included top-down components, some improvements were observed in spatial processing, temporal patterning, dichotic listening, degraded speech and SIN tests, language abilities, and reading abilities. Given that AP difficulties typically have an impact on speech perception, especially in challenging situations like degraded speech or the presence of noise, most studies aim to optimize the auditory processes underlying these difficulties. However, it could prove useful to examine in more detail the effectiveness of top-down interventions that target problem-solving as a way to proactively address communication breakdowns, language skills such as auditory closure, active listening, and attention and memory strategies. Specifically, exploring whether top-down training can enhance daily listening skills would be beneficial in understanding the practical implications of such interventions.

A secondary objective of the scoping review was to list the outcome measures used to document the effectiveness of the identified interventions. Outcome measures evaluating the effectiveness of interventions targeting AP or cognitive, language-based and learning abilities typically involved tests or test batteries specifically designed to assess the presence of conditions like an APD or a language impairment. The documented studies suggest that AP tests are the most widely used behavioural outcome measures, and there is a trend towards incorporating cognitive and language evaluations. Electrophysiological measures served as outcome variables in 13 studies. Most studies employed unique protocols, thereby posing a challenge for researchers or clinicians in selecting the most suitable protocol for effectively monitoring the intervention effectiveness. Questionnaires are another tool identified to assess the effectiveness of interventions, but they are mostly developed to screen children who might have an APD.

While one tool may excel at discriminating between individuals or classify them into categories, it may not be sensitive enough to detect meaningful changes over time [91]. In rehabilitation, measurement tools are used for two distinct purposes, namely, discrimination and detecting change, and these require different psychometric properties [91]. The current review revealed that AP behavioural measures assessing temporal ordering abilities, specifically the PPT, FPT and PPST, were the most frequently employed outcome measures. In a study by Mattsson et al. (2018), which aimed to develop normative data for AP tests with Norwegian children aged between 7 and 12 years old, temporal ordering tests, namely FPT and DPT, showed high intersubject variability, especially in the youngest participants [92]. Because of the high variance, the low scores obtained and the poor compliance on these tests, the authors did not recommend the inclusion of the FPT and DPT tests in the APD battery. Test–retest reliability was also measured with the 10-year-old participants, and was excellent for the DPT (ICC for LE = 0.83; ICC for RE = 0.82), but fair-to-good for the FPT (ICC for LE = 0.42; ICC for RE = 0.70). As a minimum of 0.70 is recommended for reliability [93], the test–retest measures for the FPT in Mattsson et al. (2018) suggest that it may not be constant or that it might lack reproducibility under unchanged test conditions. Although these results could be influenced by methodological factors (e.g., response mode and language spoken), the observed variability suggest that both the DPT and FPT may not be appropriate tools in the test battery and that the FPT might not be effective for detecting small but meaningful change.

To the knowledge of the authors, systematic reviews focusing on the psychometric properties of the wide range of available tools are lacking. This gap in the literature poses a challenge for clinicians and researchers when it comes to selecting an appropriate tool as an outcome measure. Research on the responsiveness of the various metrics is needed in order to assess whether they are sensitive enough to detect changes that are clinically significant [94,95], if they will detect the right amount of change [94], and whether they have good test–retest reliability. In addition, meaningful improvements for a participant may not be reflected in the statistical significance [95]. Assessing performance across the therapy sessions could also be of importance, as demonstrated by Jutras et al. (2015, 2019), Cameron and Dillon (2011) and Cameron et al., (2012) [45,46,50,51].

Overall, few studies included measures of the children’s perceptions of changes in their daily life-activities or listening situations. This information might facilitate determining whether the observed changes can be generalized to untrained listening situations. The studies primarily focused on exploring the parents’ perceptions rather than those of the children. The findings predominantly indicated that the interventions did not lead to significant changes in the children’s communication, i.e., their listening and auditory skills, according to parental perspectives. Furthermore, teachers were not consistently diligent in filling out and returning the questionnaires. When they did, most did not find any significant changes in the classroom environment. According to Sturgess et al.’s review (2002), children’s views can differ from those of their parents [96]. Using children’s self-reports as a tool is valid and helpful for understanding their perspectives [96].

The impacts of interventions on specific and meaningful rehabilitation goals set by the participants or their parents was not explored in any of the studies. Investigating these goals as an outcome measure could provide a clearer insight into whether the interventions had impacts on the children’s participation in life-activities that are important to them, thereby enhancing the intervention’s relevance to their needs [97]. Goal setting may also foster greater participant involvement in the rehabilitation process [97].

As discussed, only a few studies explored long-term effects of the interventions. More longitudinal studies would offer a comprehensive perspective of the long-term effects post-intervention, including whether the results are maintained and enhanced, or decline over time. Additionally, longitudinal studies would provide better insights into the maturation process of AP mechanisms in children with APD as well as their impacts in daily functioning. Furthermore, well-designed RCTs with larger cohorts are needed to better understand the potential significant impacts of interventions on children with APD. To ensure rigor, further research should include information regarding the validity and reliability of the outcome measures (e.g., responsiveness and test–retest stability). Whenever possible, measures of implementation fidelity should be completed, in addition to blinding for participants, the therapists administering the therapy, and/or the assessors.

This scoping review has limitations. The inclusion criteria were restricted to studies that reported either concurrent disorders, or, conversely, studies that specifically excluded neurological, cognitive, developmental, or congenital disorders, with the exception of learning disabilities, language impairment and ADHD. Studies that did not explicitly address the presence or absence of concurrent disorders in their participants were excluded from this review. Therefore, some studies may have been excluded due to missing participant details, even if they did not have any concurrent disorder. Additionally, if a limited number of participants in the experimental group had a developmental disorder like Asperger’s syndrome, the study was not excluded from the scoping review. Notably, no critical appraisal was conducted for the studies included in this scoping review. Consequently, there is a limitation in providing an assessment of the rigor of the methodologies employed in these studies, and readers should interpret the findings with this in mind. In some of the studies listed in the scoping review, a limitation was found in the observation that the intervention protocol and details on the activities undertaken during the intervention were not consistently and explicitly outlined.

## 5. Conclusions

This scoping review provides a comprehensive overview of intervention studies targeting AP disorder or difficulties in school-age children. A total of 42 studies, including two reviews, published between 2006 and 2022 and spanning nearly two decades, were examined. The majority of identified interventions focused on bottom-up approaches, particularly AT. A limited number of top-down interventions, primarily under multimodal approaches, were identified. A list of tools used to measure the outcomes of interventions was compiled, and the review underscores the need for research on the responsiveness of metrics assessing AP; cognitive, language-based and learning abilities; AEP and questionnaires. The scoping review also highlights the limited consideration of children’s perception of changes in daily life-activities and important listening situations. Future studies should delve deeper into parametric aspects, such as responsiveness and test–retest reliability of the outcome measures employed to assess the intervention efficacy. Conducting systematic reviews of specific interventions, such as FM systems, SIN interventions and dichotic listening interventions could offer a more comprehensive understanding of the evidence-based support they provide.

## Figures and Tables

**Figure 1 healthcare-12-01161-f001:**
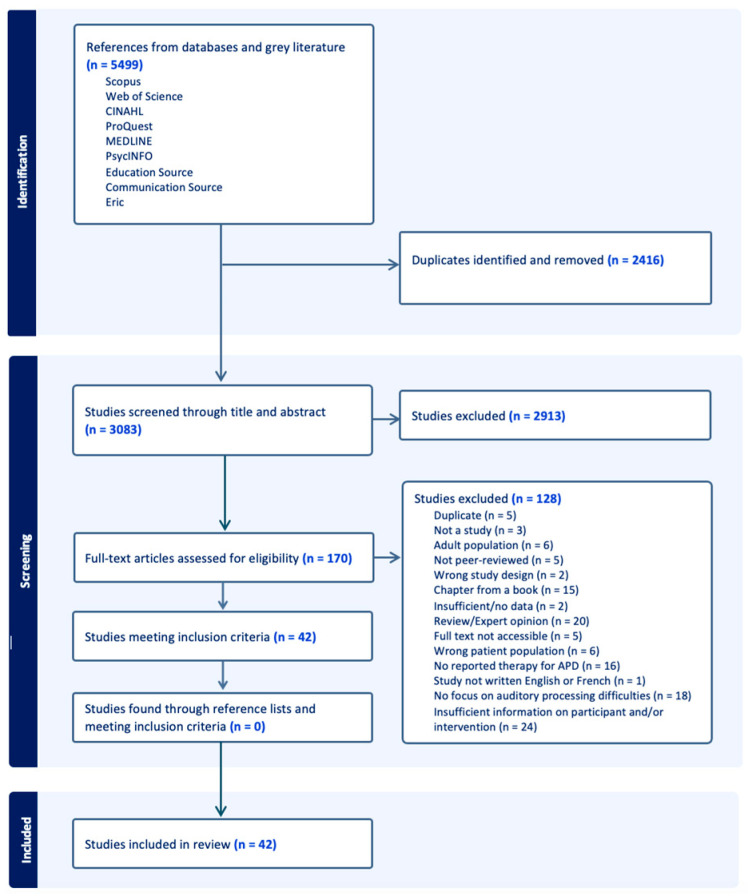
Systematic process to select relevant evidence related to interventions in children with auditory processing disorder or difficulties is illustrated in the PRISMA flow diagram.

**Table 1 healthcare-12-01161-t001:** Search strategy for Medline (Ovid), completed on 10 August 2023.

#	Search Terms and Key Words
1	exp auditory perceptual disorders/
2	(auditory adj4 perceptual adj4 (problem* or difficult* or disorder* or disease*)).ti,ab.
3	(speech adj4 noise adj4 (problem* or difficult*)).ti,ab.
4	(listening adj4 noise adj4 (problem* or difficult*)).ti,ab.
5	(Spatial adj4 processing adj4 (problem* or difficult* or disorder* or disease*)).ti,ab.
6	(auditory adj4 processing adj4 (problem* or difficult* or disorder*)).ti,ab.
7	(hearing adj4 noise adj4 (problem* or difficult*)).ti,ab.
8	1 or 2 or 3 or 4 or 5 or 6 or 7
9	exp Child/
10	(child* or kid or kids or girl or girls or boy or boys or youth* or youngster* or kindergarten* or school* or minors or p?ediatric*).ti,ab.
11	limit 8 to “child (6 to 12 years)”
12	9 or 10 or 11
13	8 and 12
14	limit 13 to (yr = “2006 -Current” and English)
15	limit 13 to (yr = “2006 -Current” and French)
16	14 OR 15
17	16 NOT (autis*).ti,ab.

**Table 2 healthcare-12-01161-t002:** Description of studies.

	Reference
** Country of study **	
Australia	[42,43,44,45,46,47,48]
Canada	[49,50,51]
Egypt	[52,53,54]
India	[55,56,57,58]
Iran	[59,60,61,62,63,64,65,66]
Israel	[67]
Brazil	[68,69,70,71,72]
Malaysia	[73]
New Zealand	[74,75]
United Kingdom	[40,41]
United States of America	[76,77,78,79,80,81]
Singapore	[38,39]
** Design **	
**1. Pre–post studies**	[46,47,48,49,50,51,52,53,54,57,59,62,67,70,71,72,73,74,75,77,78,80,81]
1.1 Repeated baseline self-control measures	[47,48,71,74,75]
1.2 Control group (no intervention)	[50,51,57,67,73,77]
1.3 Control group (with intervention)	[52,53,70,72,77]
1.4 Control group (neurotypical children)	[70]
1.5 Long-term assessment (three months to one year)	[46,47,48,50,51,73,81]
**2. Randomized controlled trials (RCTs)**	[38,39,40,41,42,44,45,55,56,58,60,64,65,66,68]
2.1 Control group (no intervention)	[40,41,55,56,58,60,64,66]
2.2 Control group (with intervention)	[38,39,42,44,45,65,68]
2.3 Control group (neurotypical children)	[42]
2.4 Long-term assessment (two weeks to three months)	[38,39,56,60,65]
**3. Non-randomized experimental study** (Two control groups: with comparative intervention and neurotypical children)	[69]
**4. Non-randomized case-controlled trial** (Control group: no intervention)	[63]
**5. Case report**	[76]
**6. Single-subject design**	[61]

## Data Availability

The data is available in each individual cited study. Otherwise, readers can access the Appendix A (i.e., Appendix A).

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
