# Peer review of "Interventions for School-Aged Children with Auditory Processing Disorder: A Scoping Review"

_healthcare, 2024, doi:10.3390/healthcare12121161_

Round 1
Reviewer 1 Report
Comments and Suggestions for Authors
Title of the manuscript: Interventions for School-aged children with auditory processing disorder: a scoping review
The present manuscript aimed to review and identify the intervention outcomes which were used for school-going children with auditory processing disorders.
1. The strength of the present systematic review is the updated information availability for the professionals like audiologists, Speech-language pathologist, and Educators dealing with these school going children.
2. This review considered both published as well as in-house literature.
3. There is a wide range of search engines used for the study.
4. The audiological management of the auditory processing disorders are listed along with the tool used for the outcome measures.
5. Most often, a systematic review excludes other systematic review and single case study. However, I can see author/s considered other systematic reviews in this study as one of the inclusion criteria. Is this not a duplication of information that is already published?
6. Page 2, line no. 92, consider deleting word “In” from the sentence In Purdy et al (2018)……,
7. Look into some the typo error throughout the manuscript.
8. Overall, study is well planned and limitation of the study is also mentioned.
I recommend the present systematic review manuscript to accept with minor revision.
Author Response
Dear reviewer,
Thank you for taking the time to review our manuscript. Your feedback is greatly appreciated.
Points 1 to 4: We agree with each point.
Point 5: Thank you for pointing this out. In our scoping review, we listed two systematic reviews (Reynolds et al., 2016 and Wilson et al., 2013). In Table S1, we summarize the inclusion/exclusion criteria for the articles as well as the general findings for each systematic review. Some of the studies included in the reported systematic reviews did not meet the inclusions criteria for our scoping review (e.g., publishing date, participant characteristics, etc). Some studies were already listed in our scoping review (i.e. Umat et al., 2011 and Sharma et al., 2012 in Reynold et al., 2013 as discussed in lines 1005-1010). Wilson et al. (2013) had only one study that met the criteria for our scoping review. The intervention by Alonso and Schochat (2009) is described in multimodal interventions of Table 3 in the scoping review and in the paragraph before the Discussion (lines 1090-1095).
Point 6: We have removed "in" as well as the comma after (2018).
Point 7: We are looking into the typos in the manuscript as well as the tables. Thank you for point this out.
Point 8: Thank you very much.
Reviewer 2 Report
Comments and Suggestions for Authors
In this scoping review, Bigras and colleagues employed a systematic search of peer-reviewed and grey literature on the auditory processing disorder (APD) from January 2006 to August 2023 across ten databases. Inclusion criteria were clearly defined following the PRSIMA protocol. They identified 42 studies predominantly focusing on auditory training interventions for APD, with limited representation of cognitive or language-based approaches. The review highlights the need for a broader range of interventions, including top-down approaches targeting cognitive and language skills. Additionally, the inclusion of outcome measures encompassing electrophysiological, cognitive, and language domains is commendable. However, the review identifies a lack of research on metric responsiveness, indicating a need for future studies to evaluate the effectiveness of interventions systematically. Overall, the scoping review provides valuable insights into interventions for APD in school-aged children. Its systematic approach to identifying interventions and outcome measures contributes to the evidence base for clinical practice. However, as the researchers pointed out, the predominance of bottom-up approaches and the limited consideration of top-down strategies warrant further investigation. Addressing these limitations could enhance the effectiveness of interventions and improve outcomes for children with APD.
I found the study well-conceived. The presentation was also very clear. I have some suggestions for a minor revision.
1. Given the lack of consensus on the definition of APD, its diagnosis can be challenging due to the variability in symptoms, comorbidity and the lack of a universally accepted diagnostic framework. APD encompasses various auditory behaviors such as discrimination, localization, and pattern recognition. APD results from a breakdown between the auditory mechanism and the brain's higher-level processing. It can manifest as global deficits, impacting language, memory, and attention. While there are standardized tests and assessment batteries available for evaluating auditory processing abilities, there is no consensus on a single set of diagnostic criteria for APD. This uncertainty would lead to variable diagnostic processes among clinicians and institutions. As the authors correctly pointed out, the American Speech-Language-Hearing Association (ASHA) and the American Academy of Audiology (AAA) have published guidelines and position statements on APD diagnosis, but these are not universally adopted. For instance, how many auditory tests should be conducted? Is the use of outlier definition for APD based on 2 or 3 standard deviations away from the mean in a single standardized auditory skill test sufficient? In addition to assessment of auditory skills, should speech-language abilities, cognitive functions, and academic performance, be included to diagnose APD? Although this review study focuses on intervention, the diagnostic procedure would presumably influence who would be included and excluded from the experimental group as well as the outcomes. The proper interpretation of the intervention outcomes would further be confounded if a randomized control design is not implemented.
2. If review articles are included, the two review articles, Bamiou et al. (2006) and Fey et al (2011), were from a decade ago. There have been several more recent articles on the topic. For instance, can neurophysiological measures be used for more objective diagnosis of APD and monitoring of intervention outcomes? You may consider including some in the introduction and discussion. Examples:
DeBonis, D. A. (2015). It is time to rethink central auditory processing disorder protocols for school-aged children. American journal of audiology, 24(2), 124-136.
Iliadou, V., Ptok, M., Grech, H., Pedersen, E. R., Brechmann, A., Deggouj, N., ... & Bamiou, D. E. (2017). A European perspective on auditory processing disorder-current knowledge and future research focus. Frontiers in neurology, 8, 622.
De Wit, E., Visser-Bochane, M. I., Steenbergen, B., Van Dijk, P., Van Der Schans, C. P., & Luinge, M. R. (2016). Characteristics of auditory processing disorders: A systematic review. Journal of Speech, Language, and Hearing Research, 59(2), 384-413.
De Wit, E., van Dijk, P., Hanekamp, S., Visser-Bochane, M. I., Steenbergen, B., van der Schans, C. P., & Luinge, M. R. (2018). Same or different: The overlap between children with auditory processing disorders and children with other developmental disorders: A systematic review. Ear and Hearing, 39(1), 1-19.
Liu, P., Zhu, H., Chen, M., Hong, Q., & Chi, X. (2021). Electrophysiological screening for children with suspected auditory processing disorder: a systematic review. Frontiers in Neurology, 12, 692840.
Drosos, K., Papanicolaou, A., Voniati, L., Panayidou, K., & Thodi, C. (2024). Auditory Processing and Speech-Sound Disorders. Brain Sciences, 14(3), 291.
3. The use of hearing assistive devices may not directly address the underlying auditory processing deficits associated with APD. Presumably, cognitive and linguistic skills can improve over time with targeted interventions and compensatory strategies. As individuals develop and mature, they may acquire coping mechanisms and adaptive strategies to mitigate the impact of APD on their daily functioning. Additionally, specialized interventions such as auditory training, speech-language therapy, and accommodations in educational or work settings can help individuals manage their symptoms more effectively. In this regard, longitudinal studies covering a wide time span is highly in need to understand how APD is influenced by various factors and the sources of variability in outcome measures. In particular, longitudinal studies are needed for documenting the natural history of APD on how the disorder manifests and evolves across different developmental stages, evaluating effectivenss and sustainable results of interventions, identifying predictors and mediators of outcomes, and assessing impact on daily functioning. Longitudinal neuroimaging studies would also help understand the underlying neurobiological mechanisms of APD and how they change over time. This can provide insights into the neural plasticity of the auditory system and inform the development of targeted interventions.
Author Response
Dear reviewer,
Thank you for reviewing our manuscript. We sincerely appreciate your time and effort invested in providing constructive comments and suggestions.
Point 1: We agree with the very important and relevant points you have made in this paragraph. The lack of consensus in identifying APD not only has an impact on the criteria used to identify APD but also on the inclusion criteria for intervention studies as well as the extent of observed changed post-intervention and, by ricochet, the generalization of the findings to the population. We added these elements in the introduction (lines 107-111 and 114-117) and discussion sections of the manuscript (lines 1104-1128).
Point 2: Following this comment, we discussed the use of neurophysiological measures for identification and as an outcome measure with what is mentioned in point 1. This information can be found in the discussion (lines 1104-1128 and lines 1176-1179).
From the list of references that you suggested, we added these references in the introduction and discussion:
1) De Wit, E., van Dijk, P., Hanekamp, S., Visser-Bochane, M. I., Steenbergen, B., van der Schans, C. P., & Luinge, M. R. (2018). Same or different: The overlap between children with auditory processing disorders and children with other developmental disorders: A systematic review. Ear and Hearing, 39(1), 1-19.
2) Iliadou, V., Ptok, M., Grech, H., Pedersen, E. R., Brechmann, A., Deggouj, N., ... & Bamiou, D. E. (2017). A European perspective on auditory processing disorder-current knowledge and future research focus. Frontiers in neurology, 8, 622.
3) Liu, P., Zhu, H., Chen, M., Hong, Q., & Chi, X. (2021). Electrophysiological screening for children with suspected auditory processing disorder: a systematic review. Frontiers in Neurology, 12, 692840.
Point 3: Following your comment, we addressed the longitudinal aspect of intervention monitoring and APD as well as RCTs in lines 1229 to 1239.
We hope that these modifications meet your expectations.
Reviewer 3 Report
Comments and Suggestions for Authors
This scoping review is on the intervention in children with auditory processing disorder. This type of manuscript is required. The authors have done a really good job in compiling the articles. I had some difficulty in following the article and I have some minor suggestions. These suggestions will improve the readability of the manuscript.
Lines 120-190: I understand the authors are trying to explain what has been done. But somehow these lines do not properly align with the need for the study. For example, in lines 183-190, what are the authors trying to convey here? They are just describing the findings of the study.
I would reorganize the last paragraph in the introduction (191-207). The authors should explain why there is a need for this scoping review.
This paragraph (lines 312-341) should be divided into different subheadings (auditory training, FM system, top-down strategies, etc…) so that it is easy for the readers to follow.
Lines 367-371: I would suggest the authors include this in table -2.
Lines 372-391: I would suggest the authors include this in a table format.
Lines 404-406: so all the studies followed ASHA (2005) guidelines for APD diagnosis?
Line 1120: hearings aids should be a separate paragraph.
Author Response
Dear reviewer,
Thank you for your valuable suggestions for improving the readability of our manuscript. We appreciate your comments.
Point 1 (Lines 120-190): Thank your for comment. We have added a short paragraph explaining these review articles. (see lines 136-140).
Point 2 (Lines 191-207): We rearranged this paragraph to make the objectives for the scoping review clearer. (see lines 204-218)
Point 3 (lines 312-341): We divided this paragraph into subheadings. It does make the paragraph easier to read. Thank you for point this out. (see lines 327-361)
Point 4 and 5 (lines 367 to 371 and 372 to 391) : See Table 2. Description of the studies (line 390).
Point 6 (404-406): It should be noted that the test batteries varied among studies and that the clinical guidelines followed were not always mentioned. This was added to line 403 to 406.
Point 7 (line 1120): Thank you for point this out. We created a new paragraph for hearing aids.
Thank you for your suggestions, and we hope that the modifications that we applied meet your expectations.
Reviewer 4 Report
Comments and Suggestions for Authors
Dear Authors,
I would like to express my appreciation for the enormous amount of work put into preparing this literature review. It is very extensive and refers to a significant number of published works in the field of research on auditory processing. The work is particularly valuable due to its thorough analysis of existing research and a clear summary and synthesis of the results. A valuable aspect of the work is the careful organization of the text, which clearly presents the goals, methods and results of the study. I also appreciate the ability to critically analyze the results, highlight their importance for clinical practice and identify areas requiring further research. The discussion would be enriched by a more detailed discussion of the methodology used in the studies included in the review, as well as consideration of any methodological limitations and flaws in these studies. Additionally, it could be beneficial to expand the discussion on the practical implications of the findings and highlight areas requiring further research to fully understand the issue. Thank you again for the opportunity to review your work. I am confident that your efforts will contribute to further progress in interventions for children with auditory processing disorders.
Kind regards,
Author Response
Dear reviewer,
Thank you for taking the time to read and offer your feedback on our manuscript. It is greatly appreciated.
Regarding this comment: "(...) a more detailed discussion of the methodology used in the studies included in the review, as well as consideration of any methodological limitations and flaws in these studies." We have added information regarding limitations of the studies and suggestions for future research (see lines 1229 to 1239 in the Discussion).
Regarding this comment: "(...) it could be beneficial to expand the discussion on the practical implications of the findings and highlight areas requiring further research to fully understand the issue." In the discussion, we highlighted the need for further research in outcome measures, but also the need to improve the procedure for identifying APD as well as its impact on understanding the effectiveness of interventions. (lines 1113 to 1128).
Thank you for your feedback and we hope that the discussion is more comprehensive.